# Influence of Guanine-Based Purines on the Oxidoreductive Reactions Involved in Normal or Altered Brain Functions

**DOI:** 10.3390/jcm12031172

**Published:** 2023-02-01

**Authors:** Mariachiara Zuccarini, Letizia Pruccoli, Martina Balducci, Patricia Giuliani, Francesco Caciagli, Renata Ciccarelli, Patrizia Di Iorio

**Affiliations:** 1Department of Medical, Oral and Biotechnological Sciences, University of Chieti-Pescara, Via dei Vestini 29, 66100 Chieti, Italy; 2Center for Advanced Studies and Technologies (CAST), University of Chieti-Pescara, Via L. Polacchi, 66100 Chieti, Italy; 3Department for Life Quality Studies, Alma Mater Studiorum-University of Bologna, 47921 Rimini, Italy

**Keywords:** reactive oxygen species (ROS), eustress and oxidative stress, memory decay, mitochondrial metabolism, nitric oxide (NO), nitric oxide synthase (NOS), guanosine (GUO), guanine (GUA)

## Abstract

The production of reactive oxygen species (ROS) in the brain is homeostatically controlled and contributes to normal neural functions. Inefficiency of control mechanisms in brain aging or pathological conditions leads to ROS overproduction with oxidative neural cell damage and degeneration. Among the compounds showing therapeutic potential against neuro-dysfunctions induced by oxidative stress are the guanine-based purines (GBPs), of which the most characterized are the nucleoside guanosine (GUO) and the nucleobase guanine (GUA), which act differently. Indeed, the administration of GUO to in vitro or in vivo models of acute brain injury (ischemia/hypoxia or trauma) or chronic neurological/neurodegenerative disorders, exerts neuroprotective and anti-inflammatory effects, decreasing the production of reactive radicals and improving mitochondrial function via multiple molecular signals. However, GUO administration to rodents also causes an amnesic effect. In contrast, the metabolite, GUA, could be effective in memory-related disorders by transiently increasing ROS production and stimulating the nitric oxide/soluble guanylate cyclase/cGMP/protein kinase G cascade, which has long been recognized as beneficial for cognitive function. Thus, it is worth pursuing further studies to ascertain the therapeutic role of GUO and GUA and to evaluate the pathological brain conditions in which these compounds could be more usefully used.

## 1. Introduction

Although the brain represents a small portion (about 2%) of the human body, it consumes more than 20% of total oxygen and glucose, which must be supplied to the neural cells together with other nutrients through a constant blood flow, as the brain has limited energy reserves [1]. Mitochondria are the main intracellular organelles which use glucose and oxygen to provide energy under the form of oxidative phosphorylation (OXPHOS)-derived ATP molecules [2]. In physiological conditions, these reactions involve the formation of a controlled amount of reactive oxygen species (ROS) and the maintenance of a cell redox homeostasis following the activity of enzymes deputed to detoxification, such as superoxide dismutase (SOD), glutathione peroxidase (GPX) and catalase. While SOD converts O2˙ to H_2_O_2_ and may diffuse to the cytoplasmic compartment, the other two enzymes convert H_2_O_2_ to H_2_O, thus leading to a balance between the production and detoxification of ROS in physiological conditions. In the nervous tissue, ROS are involved in processes such as synaptic plasticity, learning and memory and also in controlled inflammatory and immune responses [3,4,5,6]. Accordingly, these redox-related signals are indicated as oxidative “eustress”, which is fundamental for processes such as those aforementioned, occurring during brain development and adult life [7]. 

Within this framework, a paramount position is held by another gas molecule, nitric oxide (NO), which is synthesized in the brain by specific constitutive enzymes belonging to the family of nitric oxide synthases (NOSs). Neuronal NOS (nNOS) is mainly expressed in neuronal cytoplasm, but it has been also found in astrocytes and neural stem cells, whereas the endothelial isoform of NOS (eNOS) is constitutively expressed mainly in the endothelial cells. Additionally, there is an inducible form of NOS (iNOS), which is expressed upon demand in multiple cell types, including glial cells, in response to inflammatory stimuli [8]. NO is an important player in the glutamatergic transmission as well as in neurovascular regulation and therefore in oxygen supply to the cerebral tissues. Of note, NO also controls the ROS production in physiological conditions [8].

However, there are brain dysfunctions in which ROS production is no longer homeostatically controlled; for example, increases in the levels of aging ROS and/or defects in the antioxidant system may induce harmful oxidative stress [9]. In addition, in aging as well as in the pathological brain conditions reported below, ROS can rapidly react with NO, whose levels are enhanced by the increase in oxygen demand by the nervous tissue and/or as a consequence of enhanced glutamate signals. There may also be increased NO production following an abnormal activation of both nNOS and mainly glial iNOS. The reaction between ROS and NO leads to the formation of reactive nitrogen species (RNS), which have a higher diffusibility and toxicity than ROS and cooperate with them to provoke tissue/cell membranes’ damage. Indeed, ROS and RNS often cause protein and DNA oxidation as well as lipid peroxidation, which can in turn affect the mitochondrial electron chain transport and exacerbate ROS production [10]. These events may trigger a vicious cycle of oxidative stress that leads to cell death by apoptosis when a pathological threshold is passed. This threshold is typically exceeded in cases of acute brain damage such as ischemic/hypoxic events or trauma [11,12,13], and is one of the principal factors contributing to nervous tissue degeneration as in Alzheimer’s (AD), Parkinson’s (PD), Huntington’s (HD) diseases, amyotrophic lateral sclerosis (ALS) reviewed in [14] and also to the pathophysiology of other neurological disorders such as post-traumatic stress disorder, epilepsy, autism and schizophrenia [15,16,17,18]. 

Since the range of pathological situations in which ROS/RNS play a harmful role is very wide, the main question is: which mechanisms could be targeted to prevent the progression of tissue destruction after brain injury or to reverse aging? Of course, the use of antioxidants would seem the easiest answer, and many natural compounds have been proposed and tested in different pathological conditions related to ROS-induced brain dysfunctions. From an experimental point of view, the results obtained with these substances appear to be convincing, although most of them still await confirmation from a clinical/practical point of view, for example, see [19,20,21,22,23]. Among the agents so far investigated, here we want to focus on guanine-based purines (GBPs) that have been somewhat neglected. We believe that they represent a new, interesting option for the therapy of radical, related brain diseases, which are often coupled to the decline in cognitive functions such as learning and memory. In the following sections, in addition to a brief characterization of GBPs, the main protective activities of these compounds coupled to antioxidative effects in the brain will be examined. 

## 2. Outline of the Role of Guanine Base Purines in the Brain

In nature, GBPs constitute the purine system together with the corresponding adenine-based compounds. 

The family of GBPs comprises the nucleobase guanine (GUA), the nucleoside guanosine (GUO) and the nucleosides mono-, di- and triphosphate, namely GMP, GDP and GTP, respectively. Inside the cells, rather than an ex novo synthesis of purines, there is a continuous recovery and recycling of these compounds. Indeed, in physiological conditions purine nucleotides are hydrolyzed by nucleotidases to nucleosides, which are in turn broken down into ribose 1-phosphate and the corresponding bases free by the enzyme purine nucleoside phosphorylase (PNP). In turn, GUA can form GMP thanks to the activity of hypoxanthine guanine phosphoribosyl-transferase (HGPRT), which works together with specific kinases [24,25] to reconstitute the nucleotide pool (see Figure 1); if not re-used by this pathway, GUA can be converted into xanthine by guanine deaminase (GDA) and subsequently into uric acid (UA) by the enzyme xanthine oxidase [26,27]. In this way, the enzymes involved in the metabolism of purines maintain a correct balance between nucleosides, nucleotides and their deoxy derivatives, thus determining the quantity of purines released following an appropriate stimulus (see Figure 1) [28].

In the same way as the adenine-based purines (ABPs), GBPs play intracellular roles, being involved in vital cell functions. Thus, GUA is one of the known four DNA bases; more importantly, it is the most easily oxidized purine by endogenous and exogenous agents, which increases the risk of DNA damage leading to the development of cancer and other diseases [29]. GTP and GDP are crucial in the activity of GTPases linked to G protein-coupled metabotropic receptor functioning [30] as well as of small GTPases, which influence a large number of cellular processes and play important roles in neuronal functions [31]. Finally, 3′-5′-cyclic guanosine monophosphate (cGMP), formed by GMP cyclization, is a ubiquitous second messenger and it represents a key molecule in many important signaling cascades in the whole body [32,33]. 

For a long time, the effects of GBPs have been referred to as their intracellular activities. However, as previously mentioned, GBPs can be also released in the extracellular fluid, as occurs for ABPs. This aspect has mainly been characterized in the brain, as GTP is co-stored in neuronal synaptic vesicles together with ATP and co-released with the same adenine nucleotide from neurons by exocytosis [34,35]. Additionally, astrocytes are the major cerebral source of GBPs, from which these compounds are released, mainly as a consequence of ischemic conditions [36]. As observed inside cells, GBPs are metabolized in the extracellular fluid. Thus, GTP is rapidly degraded to the corresponding nucleoside GUO and then to GUA by ecto-nucleotidases and extracellular PNP, respectively [37]. Extracellular GUA can be recycled at an intracellular level by means of specific carriers for the transport of purine bases inward/outward of the cells [38]. An extracellular GDA is also active, as we recently detected in the culture medium of human neuroblastoma cells SH-SY-5Y (Figure 1) [39].

While there is a large body of literature on the extracellular ABPs acting as signal molecules in the brain, and elsewhere, e.g., [40,41,42,43,44,45,46], the articles published on the extracellular activities of GBPs are more recent and their number is limited [26,27,28,39,47,48,49,50,51,52]. This is in part due to the fact that numerous and specific receptors with their structure, downstream mechanisms and effects have so far been identified and characterized for ABPs, while this has not been the case for GBPs. Indeed, only recently our group has reported the identification of GPR23/LPA4 as a specific receptor for GUA in human tumor-cell lines [52]. Conversely, several years ago, a binding to specific sites was found for GTP in PC12 cell membranes [49] and to rat brain membranes for GUO [53]. As for this latter compound, some findings would also indicate that the effects of GUO are due to its interplay with adenosine receptors subtypes such as the A_1_ and A_2A_ receptors [54,55,56,57,58], for which GUO has been indicated as a weak agonist [55]. However, the results are not uniform [59,60] and, in any case, GTP or GUO specific receptors are still awaiting for a clear identification.

**Figure 1 jcm-12-01172-f001:**
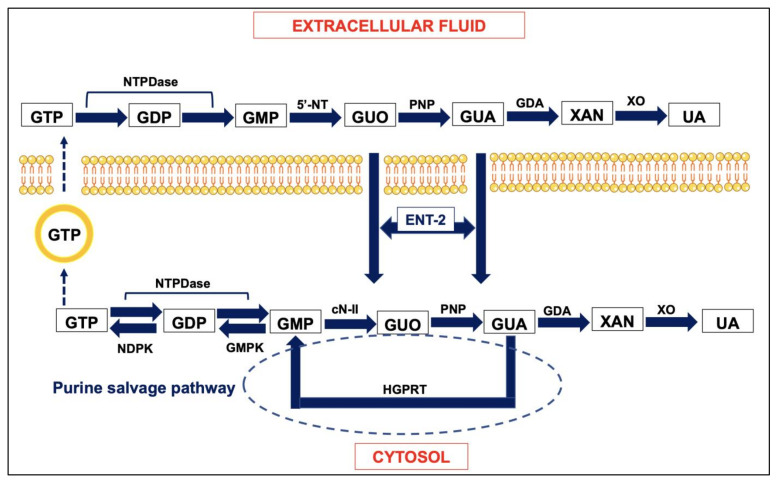
Intra- and extra-cellular metabolism of guanine-base purines. Guanosine (GUO) and GUA (guanine), which are formed in the extracellular fluid are mostly regained inside the cells by specific transporters (equilibrative nucleoside transporter 2, ENT-2) [38] and recycled in the purine salvage pathway to reconstitute the intracellular nucleotide pool thanks to the activity of specific kinases such as guanylate kinase (GMPK) and nucleotide diphosphokinase (NDPK) converting GMP into GDP and GDP into GTP, respectively [24,25]. GTP can be released from neurons by exocytosis [34,35] and in large amount from glial cells [36]. Other abbreviations: cN-II, cytosolic nucleotidase II; GDA, guanine deaminase; GTP, GDP, GMP, guanosine tri-, di-, monophosphate; HGPRT, hypoxanthine-guanine phosphoribosyl transferase; NTPDase, nucleotide triphosphate diphosphohydrolase; PNP, purine nucleoside phosphorylase; UA, uric acid; XAN, xanthine; XO, xanthine oxidase.

Nevertheless, many neuroprotective effects induced by GBPs have been reported as well as the molecular mechanisms underpinning them. Interestingly, there are differences between the GBP activities in countering some of the main pathophysiological mechanisms contributing to the onset and progression of nervous tissue dysfunction/degeneration. These differences are also related to redox signal alterations which, as previously mentioned, have been highlighted as important factors for the development/evolution of neurological and neurodegenerative diseases. These aspects will be discussed below in extensive detail. 

## 3. Neuroprotective Effects of GBPs, Mainly GUO, against Brain Oxidative Injury

There is a great deal of evidence that GBPs, and in particular GUO, exert anxiolytic, antidepressant, antinociceptive and anticonvulsant activity as well as neuroprotective effects improving repair of nervous tissue damages induced by stroke or spinal cord injury and ameliorating behavior and motor dysfunction in neurodegenerative disorders such as AD and PD, as reviewed in [26,27,49,61,62]. 

The mechanism underlying this wide range of effects was initially related to the modulation of the glutamatergic system. Indeed, glutamate plays a critical role in synaptic plasticity and stability, contributing to cerebral functions such as learning and memory. However, it may behave as a neurotoxin when it is abnormally released in the synaptic cleft after brain issue injury, inducing a cascade of events known as “excitotoxicity” [63]. A number of experimental observations demonstrated that GBPs displace glutamate binding to its receptors and also modulate glutamate uptake by neural cells reviewed in [28]. 

Subsequently, the attention was directed towards molecular mechanisms mostly activated by GUO, since it has been demonstrated that the extracellular lifetime of released GTP is short and its catabolism leads to the enhanced production of GMP and then GUO [48]. 

Thus, in addition to GUO effects on the proliferation and differentiation of nerve stem cells, which have been related to the activation of the cyclic AMP–CREB pathway [64] and can play a role in the therapy of PD [64] and depression [65], the studies have demonstrated that the main molecular pathways that participate in the neuroprotective GUO effects listed above are: (i) the class I phosphoinositide 3-kinase (PI3K) coupled to the activation of the protein kinase B (PKB), also known as Akt (PI3K/Akt); (ii) the mitogen-activated protein kinase (MAPK), originally called extracellular signal-regulated kinase (ERK); and (iii) the protein kinase C (PKC) [28,49,61]. These molecular signals are also involved in the antioxidative activity exerted by GUO, as summarized in Table 1.

Data on the main topic of this review were mostly obtained in conditions of tissue ischemia wherein the production of ROS and RNS was associated with massive deleterious effects. In particular, in astrocyte cell lines exposed to glucose deprivation, which induced, among others, an increase in ROS/RNS levels, GUO reduced the oxidative/nitrosative stress activating molecular pathways such as PKC, PI3K and MAPK/ERK that exhibited antioxidant effects (reviewed in [61]). In addition, in hippocampal slices under oxygen and glucose deprivation (OGD) condition, GUO prevented mitochondrial membrane depolarization, reducing the oxidative stress, promoting glutamate uptake and inhibiting the activity of iNOS. However, this effect, involving the MAPK/ERK pathway, required the activity of adenosine A_1_ receptor (A_1_R) and not that of adenosine A_2A_ receptors (A_2A_R) [66]. In line with some of these results, in hippocampal slices exposed to glutamate toxicity, GUO caused iNOS suppression but only at a concentration of 100 μM [67]. Conversely, in hippocampal slices subjected to OGD, GUO inhibited the increase in ROS, NO and related radical species, also preventing the loss of mitochondrial membrane potential, by inhibiting nNOS but not iNOS activity [68]. This aspect is still controversial and is waiting to be clarified. Indeed, a more recent study on hippocampal slices subjected to OGD followed by reoxygenation confirmed the modulatory effect of GUO on NO levels. This article also demonstrated that GUO, in the same way as the nonselective NOS inhibitor, N-omega-nitro-l-arginine methyl ester (L-NAME), prevented impairment in: (i) ATP production, caused by oxygen scarcity; (ii) lactate release, usually provided by astrocyte to support neurons during hypoxia; and (iii) glutamate uptake, which is aimed at limiting excessive glutamate release due to cerebral ischemia and reperfusion and the consequent excitotoxicity [69]. However, the contribution of which species of NOS are involved has not been further investigated.

Besides the modulatory effect on NOS activity, other mechanisms can account for the anti-oxidative effect induced by GUO and related compounds against brain injury. For example, glutamate-induced oxidative damage in neuroblastoma cells HT22 was counteracted by treatment of these cells with cGMP and even more by GMP and GUO. These compounds increased the levels of cysteine/glutamate antiporter system, deputed to maintain the intracellular levels of cystine for the synthesis of the detoxifying substance, glutathione [70].

Furthermore, some studies have evaluated the activity of GUO against oxidative stress caused by variations in intracellular Ca^2+^ concentrations ([Ca^2+^]i) [71]. It was found that GUO reduced the Ca^2+^-induced mitochondrial dysfunction and ROS production, contributing to restore cell metabolic and energetic homeostasis [71]. However, this finding was obtained in liver mitochondria and needs to be confirmed in cerebral tissue/cells, in which [Ca^2+^]i changes are prominent as a consequence of traumatic brain injury [72] as well as in neurodegenerative diseases [73].

Another mechanism accounting for the neuroprotective effect of GUO is linked to the interaction of this nucleoside with the inducible form of the enzyme heme-oxygenase (HO), that is heme oxygenase-1 (HO-1). This enzyme converts the pro-oxidant heme into antioxidative bilirubin and biliverdin [74]. Enhanced expression of HO-1 in the locus coeruleus has been associated with anxiolytic effects [75]. HO-1 is also the downstream protein of the nuclear factor erythroid 2-related factor 2 (Nrf2), which is activated following focal cerebral ischemia [76] and this pathway seems to be implicated in cell antioxidant defense [77,78]. Of note, while the transient induction of HO-1, as it occurs following GUO stimulation, may have a regenerative function and neurovascular protective effects, long-term HO-1 expression can cause cytotoxic effects with iron accumulation deriving from bilirubin and biliverdin metabolism [79]. The activation of the GUO/HO-1 axis has also been associated with anti-aging effects, in glial cells at least. Thus, in astrocytes derived from aged Wistar rats, GUO-induced activation of HO-1 exerted a more pronounced protective effect than in those derived from newborn animals, by inhibiting NF-kB translocation from the cytoplasm to the nucleus. This, in turn, prevented the formation of inflammatory mediators including interleukin-1 (IL-1), tumor necrosis factor-alpha (TNF-alpha), iNOS and cyclooxygenase-2 (COX2) [80]. Again, in another study on C6 rat glioma cells, GUO counteracted the oxidative/nitrosative stress induced by cell exposure to azide, a known inhibitor of mitochondrial respiratory chain, by stimulating HO-1 activity [81]. Interestingly, in another study in which neuroblastoma cell lines SH-SY5Y were subjected to blockade of mitochondrial function by rotenone plus oligomycin exposure, HO-1 involvement by GUO and the consequent mitochondrial protection were related to activation of the PI3K/Akt/GSK-3β pathway [82]. 

Finally, GUO, by interacting with the help of A_1_R and A_2A_R, was shown to modulate the SUMOylation, which is a process leading to post-translational protein modification [83,84]. Besides playing important physiological roles in the brain including synaptic maturation, regulation, and plasticity, it is a key event in neuroprotection, mainly under hypoxic/ischemic conditions [85]. Thus, it has been reported that extracellular GUO can enhance protein SUMOylation in cortical astrocytes and neurons, even though this effect was short in time, lasting only up to 1 h after GUO stimulation [83]. 

Further findings about the antioxidative properties of GUO have been obtained in in vivo models of brain damage induced by different types of injury (Table 2). For example, in adult male Wistar rats exposed to acute ammonia intoxication, intraperitoneal (i.p.) pretreatment with GUO exerted neuroprotective effects, i.e., reduced lethality and coma duration and improved electroencephalogram (EEG) traces, which were coupled to decreased levels of glutamate and alanine in the rat cerebrospinal fluid and lowered oxidative stress in the cerebral cortex [86]. Similar results were obtained in a rat model of chronic hepatic encephalopathy obtained by bile duct ligation (BDL), in which, however, GUO was i.p. administered (once a day for a week), starting 2 weeks after BDL [87]. Likewise, the antioxidative/neuroprotective activity of GUO has been shown in animal models of ischemic or traumatic brain injury. For example, GUO administration, started soon after [88] or at a maximum within 6 h [89] after focal ischemia induced by thermocoagulation in rat cortical brain, significantly reduced the infarcted area as well as the associated inflammation and neuronal degeneration, also preventing ROS production and lipid peroxidation. In both cases, these effects resulted in an improvement in rat forelimb dysfunction. Similarly, several articles in which rats suffered traumatic brain injury (TBI) have reported a protection of GUO against locomotor and behavioral impairments caused by injury, which was related to a reduction in mitochondrial dysfunction [90] and also a modulation of glutamate toxicity [91]. Interestingly, as already mentioned, GUO protection involved the activation of adenosine A_1_R [57].

While data on the antioxidative activity of GUO are relatively abundant in experimental models mimicking acute brain damage, the research has not received an analogous impulse as for GUO effects against the mitochondrial impairment coupled to increased oxidative stress observable in aging as well as in neurodegenerative and neuropsychiatric diseases [92,93,94]. So far, a few studies have evaluated this aspect. In one of these, using an in vitro model of AD such as SH-SY5Y neuroblastoma cells exposed to amyloid β (Aβ) peptide, GUO prevented cell apoptosis and ROS production [95,96], also inhibiting β-secretase activity and reducing Aβ_1–42_ peptide levels [96]. In agreement with these findings, it was more recently shown that GUO is also effective in an AD mouse model induced by intracerebroventricular injection of Aβ oligomers. Indeed, GUO, orally administered 1 h before the Aβ oligomers injection and then 1 h, 3 h and 6 h afterward, was able to cross the blood–brain barrier and recover object recognition short-term memory, as well as restoring glutamate uptake and presynaptic Ca^2+^ homeostasis with a partial protection of mitochondrial swelling [97]. Another example of the anti-oxidative properties of GUO could be inferred by the protective effect exerted by this nucleoside in an in vitro model of PD, in which 1-methyl-4-phenyl pyridinium ion (MPP^+^) was used. Indeed, this is an active metabolite of the neurotoxin 1-methyl-4-phenyl-1,2,3,6-tetrahydropyridine (MPTP) that inhibits the complex I activity in mitochondria [98]. In SH-SY5Y neuroblastoma cells, GUO was able to counteract MPP^+^-induced apoptosis [99], demonstrating, although indirectly, its capability of reducing the mitochondrial impairment and the consequent oxidative stress provoked by the toxin. Recently, it has also been observed that GUO protects rat striatal slices against oxidative damage induced by 6-hydroxydopamine (6-OHDA), a neurotoxin widely used to reproduce an animal PD model [100]. Finally, sepsis, obtained by cecal ligation and perforation in rats, is another condition in which GBPs may be effective. Indeed, this condition also causes oxidative stress in brain regions such as the hippocampus, striatum, cerebellum and cerebral cortex and GUO, acutely administered to rats, reduced lipid peroxidation, which is one of the mechanisms invoked for the GUO-induced neuroprotective effects. It is noteworthy that chronic GUO administration (10 days) also reduced cognitive impairment and depressive-like behavior [101]. A similar result was observed in rats receiving a single intravenous injection of Aβ_1–42_ peptide, in which a prolonged GUO administration (14 days) prevented memory deficits and anhedonic-like behavior [102].

**Table 1 jcm-12-01172-t001:** GUO neuroprotective activity coupled to antioxidative properties in in vitro models.

Experimental Model	Principal Mechanisms	Effect(s)	Ref.
Astrocyte cell lines under OGD	Activation of PI3K, PKC, MAPK/ERK	Reduction in oxidative stress	[61]
Hippocampal slices under OGD	Activation of MAPK/ERK in cooperation with adenosine A_1_R	Reduction in oxidative stress, promotion of Glu uptake and inhibition of nNOS	[66]
Hippocampal slices exposed to Glu	Activation of PI3K/GSK3β pathway	iNOS suppression but only at [GUO] = 100 µM	[67]
Hippocampal slices under OGD	Inhibition of nNOS activity	Mitochondria protection. Inhibition of ROS/radical species production	[68]
Hippocampal slices subjected to OGD and then reoxygenation	Inhibition of NOS activity	Prevention of the impairment of ATP production in neural cells and of lactate release and Glu uptake from astrocytes	[69]
Astrocytes from aged rats	HO-1 activation	Anti-inflammatory effects	[80]
Rat C6 glioma cells exposed to azide	HO-1 activation	Inhibition of oxidative/nitrosative stress	[81]
SH-SY5Y neuroblastoma cells subjected to mitochondrial oxidative stress with rotenone+ oligomycin Rat cortical neurons and astrocytes	HO-1 activation and involvement of PI3K/GSK3β pathway Modulation of SUMOylation (short-time effect = 1 h)	Protection against mitochondrial oxidative stress Possible neuroprotection	[82] [83]
SH-SY5Y neuroblastoma cells exposed to Aβ peptide	Inhibition of ROS production Inhibition of β-secretase	Inhibition of cell apoptosis	[95,96]
SH-SY5Y neuroblastoma cells exposed to MPP+	Involvement of PI3K pathway	Inhibition of cell apoptosis	[99]
Rat striatal slices exposed to 6-OHDA	Prevention of mitochondria dysfunction as for ATP depletion and ROS production	Protection against oxidative damage	[100]

Abbreviations: A_1_R, adenosine A_1_ receptors; Aβ, amyloid β; Glu, glutamate; GSK3β, glycogen synthase kinase-3β; iNOS, inducible nitric oxide synthase; MAPK, mitogen-activated protein kinase; ERK, extracellular signal-activated kinase; MPP+, 1-methyl-4-phenyl pyridinium ion; nNOS, neuronal nitric oxide synthase; OGD, oxygen glucose deprivation; 6-OHDA, 6-hydroxydopamine; PI3K, phosphoinositide-3 kinase; ROS, reactive oxygen species.

**Table 2 jcm-12-01172-t002:** GUO neuroprotective activity coupled to antioxidative properties in in vivo models.

Experimental Model	Principal Mechanisms	Effect(s)	Ref.
Acute ammonia intoxication in adult rats	Decreased Glu and alanine levels in CSF and oxidative stress in cerebral cortex	Reduction in lethality and coma duration; improvement of EEG traces	[86]
Chronic hepatic encephalopathy obtained by bile duct ligation in rats	Reduction in Glu and other metabolite levels in the CSF and of oxidative brain stress parameters	Attenuation of behavioral and EEG impairment	[87]
Ischemia induced by thermocoagulation in rat cortical brain	Prevention of ROS production and lipid peroxidation	Reduction in infarcted area, inflammation and neurodegeneration; improvement of forelimb dysfunction	[88,89]
Traumatic injury in rat brain induced by fluid percussion AD mouse model obtained by i.c.v. injection of Aβ oligomers	Reduction in mitochondrial dysfunction and glutamate activity	Protection against locomotor and behavioral impairments Recovery of object recognition short-term memory	[90,91]
	Involvement of adenosine A_1_R		[57]
	Restoration of glutamate uptake and pre-synaptic Ca^2+^ homeostasis; partial protection of mitochondrial swelling		[97]
Rat cecal ligation inducing oxidative stress in different brain regions	Reduction in lipid peroxidation	Neuroprotection; improvement of cognitive impairment	[101]

Abbreviations: A_1_R, adenosine A_1_ receptors; Aβ, amyloid β; CSF, cerebrospinal fluid; Glu, glutamate; EEG, electroencephalogram; i.c.v., intracerebroventricular; ROS, reactive oxygen species.

These last findings are somewhat in contrast with what has been observed in other experimental models in vivo about the chronic administration of GUO. Indeed, it has been reported that while the acute administration of GUO results in being neuroprotective, a more prolonged administration of this nucleoside can impair some cerebral functions, especially those concerning learning and memory [103,104,105]. Notably, this amnesic effect of guanosine is attributable to GUO interference with glutamatergic transmission. However, chronic administration of GUO did not impair spatial learning and memory as assessed by the Morris water maze test [49]. 

In contrast, the metabolite of GUO, that is GUA, is able to improve memory-related disorders, as discussed in the next chapter, while possible neuroprotective effects of GUA such as those exhibited by GUO have not been reported.

## 4. Neuroprotective Effects of GBPs against Learning and Memory Impairment

The impairment of cognitive function and memory retention is one of the most devastating alterations that can occur following severe or repeated cerebral ischemic events as well as in aging and many neurodegenerative diseases. A definitive treatment is not currently available, although there are many agents under intense investigation, for example, [106,107,108,109,110], mainly for therapy for the cognitive decline observed in AD and PD. 

Here, we would like to emphasize some mnesic effects of GUA exhibited in in vitro and in vivo experimental models. Ten years ago, our group observed that GUO, i.p. administered at 4 and 8 mg/kg in rat in a pretraining period, impaired memory retention following a fear-motivated avoidance task (step-down inhibitory avoidance task), normally used to assess short or long-term memory in small laboratory animals (rodents) [111]. In addition, GUO did not prevent the amnesic effect caused by the administration of 100 mg/kg L-NAME, known to reduce the capability of treated animals to acquire or retain information in learning tasks. In contrast, GUA, when administered alone at 4 and 8 mg/kg, was unable to modify the step-trough latency observed for rats submitted to the same passive avoidance task; on the contrary, when GUA was administered in the pretraining period 15 min before L-NAME, it prevented, in a dose-dependent manner, the amnesic effect of this drug. Furthermore, GUA allowed rats to also retain the memory trace when administered after training. Of note, we found that, following i.p. injection to rats, GUO was widely distributed as well as its metabolite, GUA, the levels of which rapidly increased in all tissues, including brain [112], thanks to the activity of a soluble PNP, which is present in the human plasma [113] and can also be released from glial cells [114,115]. However, GUA levels were not sufficient to hinder the GUO amnesic effect. 

To better characterize the activity of GUA, we more recently performed another study using a model in vitro of neural cells, the SH-SY5Y neuroblastoma cell line [39]. In these cells, we confirmed that GUA is present in the extracellular fluid, likely deriving from the extracellular breakdown of GUO. Additionally, we found the presence of an extracellular GDA, also known as cypin, which converts GUA into xanthine. This finding is novel and noteworthy for several reasons. First, this enzyme exerts an important activity in brain development as for dendrite pattern and spine formation [116,117]. Moreover, cypin activity can also lead to the formation of uric acid, another antioxidant with protective effect against damage caused by TBI [118], stroke [119] and spinal cord injury [120]. Notably, GDA/cypin also decreases the development of neurological disorders, such as PD [121]. 

We also explored other effects as well as some of the signal transduction pathways activated by exogenous GUA in SH-SY5Y cells. As reported in Figure 2, we found that cell exposure to this nucleobase induced a non-massive and transient ROS production without producing cell damage/death. This effect was likely due to a prevailing intracellular activity of GUA, since it was reduced by the blockade of transporters deputed to nucleoside/nucleobase reuptake. Of note, ROS production was related to the enhanced phosphorylation of some kinases belonging to the MAPK family such as apoptosis signal-regulating kinase 1 (ASK1), p38 mitogen-activated protein kinases (p38) and c-Jun N-terminal kinases (JNK). About these data, it should be emphasized that: (i) ROS are important signals in the human brain under physiological conditions [3,4,5,6,7]. In particular, ROS generation is fundamental in the generation of long-term potentiation (LTP), which has long been correlated to the formation of memory and learning functions [122]; (ii) ROS production leads to the phosphorylation/activation of some kinases, such as ASK1, JNK and p38 kinases, as we also observed in our experiments. However, the activation of this kinase loop may lead to a different outcome. While a prolonged stimulation of these kinases has been associated with apoptosis and inflammation in numerous organs, including the brain where they can contribute to neurodegenerative disorder development [123], p38 and JNK kinase phosphorylation can be important for pro-survival signals inducing autophagy, a lysosomal pathway to degrade damaged proteins, lipids and organelles [124]. 

Furthermore, in our model, GUA was able to increase the NO formation which was coupled to the phosphorylation of Akt in the PI3K pathway and to the activation of the soluble guanylate cyclase (sGC)/cGMP)/protein kinase G (PKG) downstream cascade. This finding seems to be crucial to explain the improvement of cognitive skills in rats since this pathway, also related to glutamate activity, is important in learning tasks. It is widely recognized that NO intra-/inter-cellular activity is mediated by the production of cGMP by the sGC. In turn, cGMP modulates the activity of different proteins among which a crucial role is played by PKG, a member of the serine/threonine protein kinase family. Indeed, PKG-induced phosphorylation is involved in a number of brain functions including LTP, which in turn underlies the memory formation mechanisms [125]. Moreover, this pathway is involved not only in fundamental brain functions such as synaptic transmission and memory formation, but also in neuroprotection ad immunomodulation [126,127]. 

Of note, in our hands, the activation of this pathway by GUA was pertussis-toxin-sensitive, indicating that a putative G protein-coupled receptor could be involved. This receptor could be GPR23, as we detected in some tumor cells [39]. However, this finding deserves further investigation. 

Additionally, NO production induced by GUA was reduced by cell pre-treatment with L-NAME, as it could be expected, and also by the stimulation of other molecular signals such as the Exchange Protein directly ACtivated by 3′-5′-cyclic adenosine monophosphate (EPAC-cAMP)-Calmodulin kinase II (CaMKII) pathway. While EPAC-cAMP causes [Ca^2+^]i sequestration [128], CaMKII phosphorylates nNOS, leading to a reduction in NO formation [129]. These factors are all involved in the modulation of glutamate receptor activity and thereby, in memory formation/retention [130,131]. 

Finally, in our experiments, we observed that the activation of this cascade caused the over-expression of the enzyme GDA in SH-SY5Y cells, which reduced the extracellular levels of GUA, thus modulating the duration of GUA effects. Altogether, the molecular mechanisms activated by GUA support our previous observation showing that GUA activity would improve learning and memory functions through the stimulation of NO-cGMP signaling pathway. They also corroborate the possibility that GUA administration may be useful for treating memory-related disorders.

## 5. Discussion

While a limited amount of ROS or NO in the nervous tissue seems to be useful to create a state of “eustress”, which can stimulate some cerebral functions, including synaptic plasticity and memory retention, an uncontrolled production of these molecules result in being harmful to the brain. 

Here, we tried to collect data indicating a neuroprotective activity of GBPs, in particular of GUO and GUA. From the findings reported above, it is evident that the administration of GUO or GUA can end in more appropriate results depending on the injury or dysfunction occurring in the nervous tissue. Indeed, ischemia is characterized by an acute loss of neurons consequent to high-grade inflammation and oxidative stress; this is followed by a phase in which a certain recovery of neuronal function can be observed, especially if, in the first phase of the injury, a pharmacological therapy has been set up to limit tissue damage caused by the inflammatory/oxidative stress. In contrast, in neurodegenerative and neurological diseases, etiology is mostly unknown and there is a gradual although irreversible damage of the neural cells implicating several pathogenic mechanisms including inflammation and oxidative stress. Accordingly, the type and the times of the therapeutical intervention with GBPs must be different. 

During reperfusion following ischemic events, the oxygen supplied in that period induces a greater formation of ROS/RNS which play a major pathological role. Therefore, it is not surprising that the neuroprotective effects of GUO, linked also to its anti-oxidative properties, produce the best results in ischemic-like conditions and for acute administration [88], as documented in the previous section. Of note, intranasal administration of GUO to rats with an ischemic stroke induced by thermocoagulation of pia vessels, produced prolonged neuroprotective effects even if administered up to three hours after stroke [132].

Antioxidative properties of GUO could be also useful in some neurodegenerative diseases such as AD or PD, but probably if administered in their early stages, when oxidative stress could play a major role. Indeed, as previously reported, a prolonged administration of GUO can determine an impairment of memory retention, which could worsen the course of those diseases and also impair the post-ischemic cognitive recovery. 

On the other hand, GUA, which per se is not provided with anti-oxidative properties, should be more appropriate in those pathological conditions in which memory decay is one of the main handicaps. Indeed, GUA could lead to the activation of GDA. This enzyme seems to exert neuroprotective effects, also causing the formation of antioxidants such as UA. More importantly, GUA, before being metabolized, could activate the cGMP/NO/PKG pathway which has long been demonstrated as able to prevent/reduce memory impairment [116,117,118]. Therefore, GUA should be more useful in pathological conditions in which neuronal loss of function occurs slowly and is coupled to memory decay. However, while the mnesic activity of GUA has been characterized, both in animal and in vitro models, the involvement of GDA as protective anti-oxidative molecule still needs to be better characterized. 

In conclusion, GUO and GUA are two agents exerting important although different effects in many brain dysfunctions. However, the road to their potential therapeutical use is still long. Firstly, there is the need to identify possible receptor sites for GUO and GUA in the brain, also confirming the interaction of the latter with the GPR23 receptor in neural cells. Moreover, it should be mentioned that GUO is sparingly soluble in water and therefore is relatively diffusible in the body fluids, while GUA is almost insoluble in water but can be solubilized in acidulated water or in a solution of potassium hydroxide solution. Finally, possible side effects consequent to GUO or GUA administration have been poorly investigated so far. Therefore, before going to the clinical phase, there is a lot of experimental work to be completed, as well as solving the pharmacokinetics issues related to drug administration. 

## Figures and Tables

**Figure 2 jcm-12-01172-f002:**
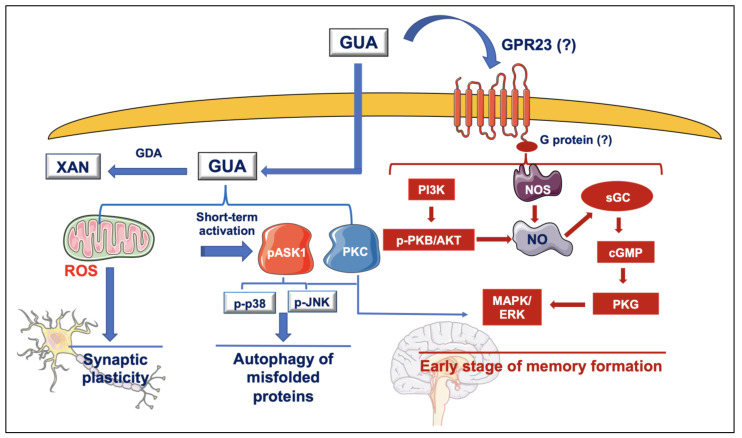
Scheme of the molecular pathways activated by SH-SY5Y neuroblastoma cell exposure to GUA, as reported in [39]. Abbreviations: cGMP, cyclic GMP; GDA, guanine deaminase; GUA, guanine; MAPK/ERK, mitogen-activated protein kinase/extracellular signal-regulated kinase; NO, nitric oxide; NOS, nitric oxide synthase; p-ASK1, phosphorylated apoptosis signal-regulating kinase 1; p-JNK, phosphorylated c-Jun N-terminal kinase; p-p38, phosphorylated p38 kinase; p-PKB, phosphorylated protein kinase B; PI3K, phosphoinositide-3 kinase; PKC, protein kinase C; PKG, protein kinase G; ROS, reactive oxygen species; sGC, soluble guanylate cyclase; XAN, xanthine.

## Data Availability

Not applicable.

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
