# Peer review of "Influence of Guanine-Based Purines on the Oxidoreductive Reactions Involved in Normal or Altered Brain Functions"

_jcm, 2023, doi:10.3390/jcm12031172_

Round 1

Reviewer 1 Report

-         

       The authors have written the review very well and the review provides up-to-date information about the role of GBPs in various brain functions. The authors successfully summarized the evidence supporting the information related to normal brain function as well as during neurodegenerative disorders. I have a few comments and suggestions before the manuscript is accepted for publication. 

       1. Line 55: position is “held” instead of “hold”

-         2. Table 1: SH-SY5Y and SHSY-5Y. Use constant naming

-          3. I think there is very less information from “in vivo” data, especially under heading 3. It would be nice to include more data from rodents or similar experiments, if available.

-          4. Under heading number 3, there is a lot of scattered information. It would be nice if the authors include a comprehensive figure for the whole information or at least make sub-headings to increase the readability of the paper. 

-          Overall, there are some typos and minor grammatical errors which need to be corrected. 

Author Response

Answers to the comments of this Reviewer, whom we thank very much for his/her help 

  1. Line 55: position is “held” instead of “hold”

R.: we have corrected the indicated verb.

  1. Table 1: SH-SY5Y and SHSY-5Y. Use constant naming

R.: we apologize for this mistake and have revised the text according to the suggestion by this Reviewer.

  1. I think there is very less information from “in vivo” data, especially under heading 3. It would be nice to include more data from rodents or similar experiments, if available.

R.: we thank this Reviewer for his/her inputs. Accordingly, we have added a series of information on the data obtained with the use of GUO in “in vivo” models, as witnessed by the new references included in the bibliography.

  1. Under heading number 3, there is a lot of scattered information. It would be nice if the authors include a comprehensive figure for the whole information or at least make sub-headings to increase the readability of the paper. 

R.: we distributed data obtained by the use of GUO in vitro and in vivo in Table 1 and 2, respectively. By this way, we think that the readability of our manuscript is improved.

  1. Overall, there are some typos and minor grammatical errors which need to be corrected. 

R.: we have carefully reviewed the manuscript trying to do our best and to correct all errors.

All changes are highlighted in yellow.

Reviewer 2 Report

The authors gave the detailed overview about effects of guanine-based compounds (GBPs) against neurodysfunctions induced by oxidative stress. The review is of sufficient significance and originality, however some abbreviations need to be introduced/defined (like TBI, JUNK, etc), while others need to be used in the same manner throughout the manuscript (for example P38, etc), and some terms are defined several times (for instance long-term potentiation (LTP), etc). The authors should read the manuscript thoroughly and uniform these items.

Author Response

Answer to the Reviewer #2

The authors gave the detailed overview about effects of guanine-based compounds (GBPs) against neuro-dysfunctions induced by oxidative stress. The review is of sufficient significance and originality, however some abbreviations need to be introduced/defined (like TBI, JUNK, etc), while others need to be used in the same manner throughout the manuscript (for example P38, etc), and some terms are defined several times (for instance long-term potentiation (LTP), etc). The authors should read the manuscript thoroughly and uniform these items.

R.: we thank this Reviewer for his/her appreciation of our work. We have reviewed the manuscript seeking to correct the errors indicated.

All changes are highlighted in yellow.